# COVID-19 press conference search engine using BERT

## Abstract

There have been multiple press conferences concerning COVID-19, where governments present their efforts in fighting the pandemic. These briefings provide reporters with a platform for their questions to be answered. This work studies multiple press conferences from different governments and agencies, ranging from WHO to the Whitehouse to different state governors to even different governments. This work collects the transcripts of these press conferences, then using a custom heuristic, selects short exchanges between different speakers, hence selecting exchanges made by the reporters. Then using a custom trained sentence-classifier, selects the questions raised by the reporters through these exchanges. This creates a new dataset, which contains the questions asked by reporters and how they were answered by officials. This dataset can prove useful in a number of applications, in this work we present one of these uses, which is building a search engine. This search engine is built on these questions by fine-tuning the state-of-the-art BERT language model on the collected COVID-19 press conference transcript dataset. This search engine can prove helpful in answering questions raised by the public and knowing how they were answered by officials, it can also help reporters and researchers in finding how a specific question was answered by the different governments. Our goal by this work is to help organize the press questions concerning COVID-19 to help build an insight on the different efforts being taken to combat the pandemic.

## 1 Introduction

Press conferences are the channel of communication that governments/agencies use to communicate their efforts in fighting COVID-19 with the world. Studying and analyzing the transcripts of these conferences would provide great insights on the different approaches and efforts these governments/agencies use in their fight against the pandemic.

This work aims to collect the transcript of multiple press conferences made since late January, made by different governments/agencies. Then using a custom heuristic, the short exchanges made throughout these conferences are selected, which mostly contains the exchanges made by reporters. A sentence-classifier (a CNN model trained on a combination of SQuAD [1] (Rajpurkar et al., 2018) and SPAADIA [2] datasets) is then used to only select the questions raised by the reporters from these exchanges. This builds a dataset containing questions raised by reporters and how they were answered by the officials from different governments. This dataset can prove useful in a number of applications, from building an insight on the questions, analyzing when a certain type of question was mostly raised, comparing the questions raised from reporters to different governments. In our work we introduce another application of this dataset, which is building a customized search engine capable of finding the most similar question to a custom query. This can prove helpful in answering questions raised by the public, it can also help reporters build an insight on how a specific question was answered by the different governments.

This search engine is built by fine-tuning BERT (Devlin et al., 2018), a state-of-the-art language modeling, to build a customized language model, capable of understanding the context of COVID-19 press conferences. An evaluation was built on BERT to test how well it understood the context of the COVID-Press dataset. We use the evaluation technique proposed by (Ein Dor et al., 2018), which tests the ability of BERT to identity sim-

---

[1] https://rajpurkar.github.io/SQuAD-explorer/
[2] http://martinweisser.org/index.html#Amex_a

ilar sentences, we have built our own similarity dataset from COVID-19 press context for evaluation. Then using the recent proposed architecture SBERT (Reimers and Gurevych, 2019) (which builds a mechanism for selecting the most optimized embedding for sentences from BERT), a search engine is built. This search engine gets the most similar questions and their answers (from the built dataset) to a user query.

The paper is structured in the following way: (Section 2) presents how the dataset was collected and the proposed method for selecting the questions and answers in a press conference. In (section 3), we view how BERT (Devlin et al., 2018) was fine-tuned to the collected dataset. (Section 4) views the used architecture for extracting the embeddings from the fine-tuned BERT using the newly proposed SBERT (Reimers and Gurevych, 2019). (Section 5) views some results on running the search engine. We have used google colab for scrapping, fine-tuning and building our search engine, the code [3] is provided as jupyter notebooks to run seamlessly on google colab. The data [4] is hosted on google drive to connect seamlessly with google colab.

This work opens the opportunity to analyze how a certain question is answered across the different governments, hence building insights on the different fighting efforts being made across the world.

## 2 Building COVID-19 Press Dataset

This work uses the transcripts provided by **REV**

```
https://www.rev.com/
blog/transcript-tag/
coronavirus-update-transcripts
```

**REV** provides the transcripts of the press conferences made by multiple governments and agencies, these are :

- World Health organization press briefings

- United Kingdom Coronavirus briefing

- White house press conferences

- Justin Trudeau Canada COVID-19 Press Conference

---

[3]https://github.com/theamrzaki/covid-19-press-briefings
[4]https://github.com/theamrzaki/covid-19-press-briefings#data

- Press Conferences made by multiple US state governors (NewYork, Iowa, Florida, ... and many others)

### 2.1 Scrapping Transcripts

We have built a customized scrapper using python to scrape the exchanges made by different speakers in a given press conference. We have scrapped 654 press conferences made since 23th January, till 12th May. We were able to obtain more than 66k exchanges throughout the collected transcripts dataset. We tend to scrape the transcript text of each speaker, with the name of that speaker, with the timing of when this exchange was spoken within the press conference. We also record the name and the date of the press conference in addition to its url (from **REV**).

Since COVID-19 is a continuously evolving situation, we would periodically run our scrappers to obtain the most up-to-date transcripts.

### 2.2 Building COVID-19 questions dataset

This work aims to build a search engine on the questions raised by reporters and how they were answered by officials. However selecting these exchanges from the scrapped dataset appeared quite challenging, as **REV** doesn't provide a guide on the identity of each speaker, so work must be done in order to try and identify the identity of each speaker.

To select the questions raised by reporters, our work was broken down into 2 steps. First selecting all the reporter exchanges, then selecting the questions from these exchanges. We first build a custom heuristic capable of identifying the exchanges made the reporters, then we build a custom sentence-classifier to select the raised questions.

#### 2.2.1 Custom heuristic for selecting short exchanges

This work uses a custom heuristic to try and identify the identity of the speakers, in order to select the exchanges made the reporters. This heuristic is built over rules of when the speakers begin to speak and the amount their exchanges. The proposed rules are:

- The longest exchanges in a press conference are flagged to be spoken by the official giving the press conference (president, prime minister, governor or a health official).

- The first exchange, is flagged as been spoken by the presenter (the conductor of the conference). This can either be a reporter or the official himself.

- If the main official conducting the conference, mentioned other speakers, those speakers are flagged to be helpers to that official. In most cases these have been found to be either health officials (like in case of Dr Fauci in the white house conferences), or other officials (either military or a financial official).

- We are most concerned with flagging the reporter exchanges. These have been found to be few exchanges in a single press conference made by each reporter (each speaker speaks either once or twice max). When this pattern is found (few exchanges made by a single speaker), these exchanges are flagged to be made by reporters, and are considered to be questions. Then the exchange right after it is flagged as its answer.

Using these rules, the previously collected transcript dataset was flagged with the proposed speakers (either conference-conductor, official, helper, or reporter). A dataset is then built to only contain the exchanges made by the reporters and the answers to them. However, not all of the selected reporter exchanges can be considered as questions, this is why a custom sentence-classifier has been built in order to only select the questions.

### 2.2.2 Sentence-Classifier for selecting questions

A classifier was built with the goal of correctly identifying the true questions from the built reporter exchanges dataset.

We used the model presented by [5] which builds a CNN model on a combination of SQuAD [6] (Rajpurkar et al., 2018) and SPAADIA [7] datasets. These datasets classify sentences into 3 classes

1. 1111 Command

2. 80167 Statement

3. 131001 Question

---

[5] https://github.com/lettergram/sentence-classification
[6] https://rajpurkar.github.io/SQuAD-explorer/
[7] http://martinweisser.org/index.html#Amex_a

In our work we are only interested in classifying questions, so we have considered both the "Command" and the "Statement" as the same class.

The CNN model was trained on 170077 sentence of the 3 classes, then it was tested on 42520 sentence. It was able to achieve a test accuracy of 0.9948.

This model has then been used to classify the questions from the collected reporter exchanges. It classified that 67.76% were indeed questions. These correctly identified as questions (about 5k) where then selected (with their answers) in a new dataset which only contain the reporter questions.

## 3 Fine-Tuning BERT to COVID-19-press

BERT (Devlin et al., 2018) has been proven to be the state-of-art architecture for language modeling. It is built as an enhancement to the vanilla transformer (Vaswani et al., 2017). It is built to only contain an encoder structure, and to depend solely on self-attention.

BERT is unique in the approach used in its pre-training, where "masked language model" (MLM) is used as the pre-training objective, inspired by the Cloze task (Taylor, 1953). This approach randomly chooses words from the input text (15% of words), and the training objective is to predict these masked words. This training objective enables BERT to be pre-trained in an unsupervised manner, where raw text is supplied to BERT, without having labels.

This training objective is also used in its fine-tuning, in our case, the collected dataset (COVID-19 press of 66k exchanges) is used as the raw training text to fine-tune the pretrained BERT. Hugging Face (https://huggingface.co/) library was used to fine-tune BERT to the collected dataset. The BERT model provided by google (https://huggingface.co/google/bert_uncased_L-8_H-256_A-4) was used as our pre-trained BERT. Google colab was used as the platform for fine-tuning.

### 3.1 BERT Evaluation for COVID-19 press context

Evaluation of a customized language model proves challenging, as most of the available evaluation techniques are build to cope with a general language model not a customized one. A recent evaluation technique for BERT was recently proposed by (Ein Dor et al., 2018). This technique relies on evaluating how BERT is able to measure the

similarity of different sentences, where 3 sentences are supplied to BERT, 2 are similar and 1 is not. The evaluation is made to test if BERT is able to correctly identify the similar sentences. The true breakthrough that this technique offers over other evaluation mechanisms, is the ease of producing customized evaluation datasets without manual labeling.

In (Ein Dor et al., 2018) they were able to create a customized similarity dataset from scrapping Wikipedia pages. They assumed that sentences from the same paragraph in a Wikipedia article are similar, and a sentence from a different paragraph would talk about a different subject, hence lower similarity. They then used this to build a customized similarity dataset by using a Wikipedia article of their chosen context.

In this work, we have used the same approach to build our own similarity dataset. We have used our built dataset, that contain all the exchanges between speakers of all datasets (dataset of 66k exchange). Then to build the similarity dataset, we selected every 2 adjacent exchanges from a press conference as the similar sentences, we then used an exchange from a different press conference as the different sentence. By this, a dataset of 40k triplets has been created, where each row contains 3 sentences, 2 similar (of the same press conference), and one different.

BERT has been evaluated using this custom built evaluation dataset, it was capable of correctly identifying 99.7% of the 40k triplets. This indicates that BERT was capable to correctly understand thee context of the sentences. We also evaluated our fine-tuned version of BERT, where it scored an accuracy of 99.88%, which indicates that even the examples which were quite difficult for the vanilla BERT to correctly identify, were correctly handled by our fine-tuned BERT.

## 4   BERT to build a search engine

Using BERT for sentence-pair regression (measuring how similar sentences are to each other, the technique used to built a search engine), proves to be inefficient for multiple reasons.

To begin with, for sentence-pair regression in BERT, the 2 sentences are provided to BERT with a special separator token in between them [SEP]. To build a search engine using this approach, one would need to supply each sentence to BERT (in addition to the query sentence). This would require running BERT each time in deployment for about 5k times (size of the dataset) to get the most similar question and its answer from all the dataset. This is simply unsuitable for building a search engine.

Another approach other than sentence-pair regression is often proposed, which is extracting the sentence embedding from BERT. First running BERT just once on the 5k questions, getting their embedding, and in deployment, just run BERT once on the query and use cosine similarity to get the most similar question and its answer. However this also exposes another disadvantage in BERT, as in BERT no independent sentence-embedding are computed, this makes it challenging to extract a good embedding from BERT (Reimers and Gurevych, 2019).

Multiple approaches were proposed to help extract good embeddings from BERT. ((May et al., 2019),(Zhang et al., 2019),(Qiao et al., 2019)) proposed using the [CLS] token from BERT as the fixed size vector embedding for a sentence. Another approach used by (Reimers and Gurevych, 2019), computes the mean of all output vectors.

In (Reimers and Gurevych, 2019), they trained a Siamese BERT network on SNLI data (Bowman et al., 2015) and on Multi-Genre NLI. They then evaluated different polling approaches to build embedding representation for sentences. Either using [CLS] or by averaging vectors to get [MEAN], they fine-tuned their architecture for classification objective function on the STS benchmark with regression objective function. They concluded that using the [MEAN] polling strategy outperformed that of using [CLS] strategy. This is the reason it was the selected pooling strategy in our work.

## 5   Experiments

To build our search engine, we fine-tuned BERT on the collected 5k questions, saved their embedding using the [MEAN] polling strategy, then for each test query, we run the fine-tuned BERT with the same polling strategy, and using cosine similarity we get the most similar questions asked in the collected press-conferences.

To select the test queries, we followed a selecting mechanism to automatically select sentences from our corpus. Some measures were taken to ensure that the selected sentences were of different context. The resultant embedding from the fine-tuned BERT were used with k-means to cluster the dataset to multiple clusters, were each of them

convey a specific context. Elbow method was used to identity that 10 clusters would be the optimized number of clusters to be used (dataset with clusters as labels). The clusters with the most number of associated sentences were then selected to draw the test sentences from, then using a random generator, a sentence from each cluster was selected.

The following are some examples from the search engine, the top 2 most similar questions and their answers are selected. With the name of the press briefing, its date, and the time within the briefing when this exchange was spoken.

---

### Input Sentence

And regarding unemployment, we're hearing stories of people are still not getting returned phone calls within 72 hours,...

---

### Results

**Score**: 0.9296

**question**: Yes, governor, I want to go back to unemployment. We're still hearing from many who are wanting to know when they're going to get their checks and you gave that answer,...

**answers**: Yeah, I think that is right. We have processed and I think most of the checks that are direct deposited have gone out to I think a majority of the people who are in that backlog....

**header**: Transcript: Governor Ned Lamont COVID-19 Press Conference Transcript April 14

**date**: Apr 14, 2020 (39:31)

---

**Score**: 0.9252

**question**: I'm still healing hearing from some people who are having problems getting through to unemployment and getting the benefits that they feel they're entitled to.....

**answers**: So I think that it's always important to have some perspective here. We've had over a million people become unemployed in the last six weeks. We have been able to make sure that over 820,000 people have gotten the assistance that they've earned...

**header**: Michigan Governor Gretchen Whitmer Press Conference Transcript April 24

**date**: Apr 24, 2020 (30:53)

---

Table 1: Query 1

---

### Input Sentence

This morning at the San Mateo county board of supervisors meeting officials there expressed grave concern about the lack of PPE at Seton Medical Center, and also the need for more staffing. I just want to find out what the state is doing to address those needs?

---

### Results

**Score**: 0.9039

**question**: Reporters in the room, I'm working on behalf of your colleagues. I'm going to try and get some of their other questions in.We may not have as many confirmed cases downstate but already clusters of cases in a senior home in Taylorville outnumber the available number of ICU beds at the hospital in town....

**answers**: Our ICU bed situation in the state, as you know this is as we move toward the peak of this, we are going to be filling up ICU beds across the state. It isn't the same in every area. There are critical-access hospitals that may have fewer ICU beds. There are other hospitals in other areas of the state that may have more availability,....

**header**: Illinois Governor J.B. Pritzker COVID-19 Briefing Transcript April 1

**date**: Apr 1, 2020 (40:52)

---

**Score**: 0.9038

**question**: The next question is for the Secretary. Dr. Levine from the Capitol Star. HAP said that it was in talks with the administration today about resuming non-emergent services as the lockdown eases. Can you characterize the state of those talks and what you would need to do to allow hospitals to start treating those patients?

**answers**: Mm-hmm (affirmative). So that is correct. We have had discussions with the hospital association as well as a number of different health systems and hospitals about when would be the right time to allow non-emergent procedures to occur. Now remember, some of those are procedures that really have to happen for people's health and they've been on hold and it's really difficult.

**header**: Pennsylvania Gov. Tom Wolf Coronavirus Briefing Transcript April 22

**date**: Apr 22, 2020 (19:00)

---

Table 2: Query 2

As seen in the previous examples, the exchanges that were flagged as questions were indeed questions. This helps indicate that the used mechanisms for selecting questions from the different exchanges were successful.

## 6 Conclusions

In this work we present a new COVID-19 data source, which is the press conference briefings, as a rich source for analyzing different governments response for fighting the virus. We also present some mechanisms of selecting questions from these press briefings. We have used the state-of-art language models for building a semantic search engine to get the most similar questions from the press briefings. This search engine can prove helpful in addressing the questions posed by the public concerning COVID-19. It can also be used by journalists and researchers in comparing the different efforts made by the governments around the world in fighting the pandemic.

Building a search engine is just one of multiple possible applications of using this dataset. Further analysis of this dataset opens the possibility to multiple other uses, like analyzing the timeline of asking a certain question, when it was first raised, by whom and how it was answered.

We believe that this new data source can prove useful in multiple areas of research, to understand and build insights on the different approaches taken by governments in combating this virus.

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
