# OpenReview forum: "COVID-19 press conference search engine using BERT"
_aclweb.org/ACL/2020/Workshop/NLP-COVID — Submitted to NLP-COVID-2020_

### Official Review · AnonReviewer3 · 2020-06-16
**No innovation, no proper evaluation**

**Rating:** 2
**Confidence:** 4

**Review:**

The paper reports on the implementation of a search system over press conferences related to COVID-19. The search system retrieves the most similar questions to a user query, and their answers.

The data set is built by scraping the press conferences. The system uses simple heuristics to determine whether the speaker is a reporter asking a question or an answer to a reporter's question.

The search system uses cosine similarity between the embedding of the query and the candidate question. Embeddings are found by fine-tuning BERT.

The pros of this paper are:

1. The paper presents a system that can be useful to search through press-conferences.

2. The code is available in github (I have not looked at it to avoid concerns with anonymity).

The cons of this paper are:

1. Even though  the system is based on some of the latest technology, there is no real innovation in this paper.

2. There is no proper evaluation. Instead, a few examples have been included in the paper. The paper did not say how these examples have been selected and they could have been cherry-picked to show the best outputs.

3. There is no error analysis or discussion of how the system could be improved.

Below are some additional comments.

- There are several language errors in the paper but the paper is understandable.

- It would be useful so see some examples of the collected transcripts, perhaps in an appendix.

- The paper presents some simple heuristics to identify the questions asked by reporters and their answers. Have these been evaluated? What is the quality of the resulting resource?

- Give more details about how the questions for your experiments were chosen. Were they real questions asked by someone? Where they constructed by the authors? If they were constructed by the authors, what criteria did they follow to avoid a bias in the choice of questions?

---

### Official Review · AnonReviewer2 · 2020-06-18
**Searching press conference transcripts for Covid-19**

**Rating:** 3
**Confidence:** 5

**Review:**

The Covid-19 pandemic has created a wealth of information sources that can be processed to provide value to those needing more information on the global crisis. This paper uses a collection of press conferences to build a corpus of questions and their answers. The authors process the corpus to develop a source of questions and answers that have potential uses.

Being able to search through press conferences, and the questions asked and answered, could be a valuable task. However, the work presented in this paper is way too preliminary. It is not clear exactly what the authors propose to do with the corpus or how it will be evaluated.

They also need to spell the word “scrape” correctly, especially since scraping this Web site is so central to their work.

But more importantly, they need to describe in more detail the task for which they are building their system. They then need to define proper evaluation measures for their task, and create a test collection that allows such an evaluation.

This could be the beginning of some interesting work from what looks to be a valuable corpus. With additional work, this could be a valuable contribution.

---

### Comment · AnonReviewer1 · 2020-06-18
**Potentially interesting, but not ready for publication yet.**

I agree with all the previous reviews. Instead of repeating feedback
they've already given, I'll attempt to be more constructive and
envision where this work can go.

The authors propose a new task, that of searching press conferences.
Every new task, however, needs to begin with a model of users and
their information needs. Who's going to use this search engine, and
why would they want to search press conferences?

The authors mention this briefly at the end of the introduction, but
there needs to be significantly more elaboration. I can think of at
least two plausible scenarios:

+ Citizens searching for advice from government officials, e.g.,
  Should I be wearing a mask?

+ Journalists investigating differences across time (e.g., how has
  advice on mask wearing evolved over time, in the same jurisdiction)
  and/or across different jurisdictions (e.g., how do mask wearing
  mandates compare across countries).

The answer to this question will shape the rest of the work. Without a
clear answer, talk about data preparation, neural models, and formal
evaluations (in this case, the lack thereof) is pre-mature. If you
don't know what users may want, how do you know what to build? And how
can you evaluate what you've built?

In technical terms, the system that the authors describe is,
essentially, a FAQ retrieval system, with question/answer pairs
extracted from press conference transcripts with heuristics. The
authors should ground their techniques in the vast literature in this
space; the closest today goes under the banner of "community QA"
systems.

In terms of modeling, it is unclear that you need neural models, at
least for the few sample questions that were provided --- there's
quite a large overlap in terms of keywords. I'd like to see a keyword
search baseline as part of any formal evaluations.

In summary, I think there's potential here for this work to go
somewhere interesting. However, in its present state, the paper is not
ready for publication.

---

### Author Response · Authors · 2020-06-21
**Modifications**

We would like to thank you all for your comments.
We have built some modifications to the paper taking into consideration your review.

1- Ensuring the correct selection of questions to build the QA dataset :
we have followed 2 steps in building our dataset, first, we used the same heuristic measures (of the last submission) to identify the speakers, where we select the exchanges made by the reporters. Then the second step which was added in this revision, ensures that only the questions are been selected, where a custom-built sentence-classifier (CNN model trained on a combination of SQuAD & SPAADIA datasets) was used to only select the question from all the different exchanges.

2- Evaluation of BERT :
We used an evaluation technique recently proposed to evaluate how well BERT models understand a given context. It tests how well BERT can extract the similarity between different sentences. A custom similarity dataset is built from the collected dataset, where each row contains 3 sentences (2 similar 1 not), and BERT is tested by how it can correctly identify similar sentences. This evaluation was made on BERT before and after fine-tuning. The results show how well the BERT models were capable of understanding the given COVID-19 press context.

3- Selection of test sentences.:
To select the test sentence, we have reported the used mechanism. Our goal is to automatically select sentences with a different context.
We used the resultant embedding from BERT with k-means to cluster the context into 10 clusters (ELbow method was used to get that 10 clusters were the optimized number of clusters). Then the clusters with the most number of sentences associated with it were used, then a random sentence from each of these selected clusters was drawn as our test sentence.

4- Search Engine as one of multiple other potential uses of this dataset :
We have reported that building a search engine is just one of multiple other potential uses of this dataset. We have pointed out how this search engine could prove useful. We have also reported some other potential applications for this dataset.

5- Availablity of the dataset, while ensuring anonymity :
To ensure the anonymity of submission, we have uploaded our datasets to dropbox, where it won't show the identity of the uploader
https://www.dropbox.com/sh/qri45sfy5vlsesj/AACBI3REr_HNG5FVcpdsCcvDa

6- Language mistakes :
we have corrected multiple language mistakes in the paper

We truly like to know your reviews of this new revision

---

### Decision · Program_Chairs · 2020-06-18

**Decision:**

Reject

**Comment:**

Based on the reviewer feedback, the decision is to Reject this paper.

The reviewers have identified that the paper is heading in a potentially interesting direction, but that it isn't sufficiently mature at this stage to be accepted for publication.

Thank you for your submission, and we hope that you find the feedback valuable for the future development of this work.